# Targeting Ferroptosis Holds Potential for Intervertebral Disc Degeneration Therapy

**DOI:** 10.3390/cells11213508

**Published:** 2022-11-05

**Authors:** Jiaxing Chen, Xinyu Yang, Yi Feng, Qiaochu Li, Jingjin Ma, Linbang Wang, Zhengxue Quan

**Affiliations:** 1Department of Orthopedics, The First Affiliated Hospital of Chongqing Medical University, Chongqing 400016, China; 2Orthopedic Laboratory, Chongqing Medical University, Chongqing 400016, China; 3Medical Education Department, The First Affiliated Hospital of Chongqing Medical University, Chongqing 400016, China; 4Department of Orthopedics, Peking University Third Hospital, Beijing 100191, China

**Keywords:** intervertebral disc degeneration, ferroptosis, iron, lipid peroxidation, treatment

## Abstract

Intervertebral disc degeneration (IVDD) is a common pathological condition responsible for lower back pain, which can significantly increase economic and social burdens. Although considerable efforts have been made to identify potential mechanisms of disc degeneration, the treatment of IVDD is not satisfactory. Ferroptosis, a recently reported form of regulated cell death (RCD), is characterized by iron-dependent lipid peroxidation and has been demonstrated to be responsible for a variety of degenerative diseases. Accumulating evidence suggests that ferroptosis is implicated in IVDD by decreasing viability and increasing extracellular matrix degradation of nucleus pulposus cells, annulus fibrosus cells, or endplate chondrocytes. In this review, we summarize the literature regarding ferroptosis of intervertebral disc cells and discuss its molecular pathways and biomarkers for treating IVDD. Importantly, ferroptosis is verified as a promising therapeutic target for IVDD.

## 1. Introduction

Intervertebral disc degeneration (IVDD) is a common aging disease among humans [1]. Nearly 80% of adults are impacted at some point during their life, and 10% of patients with IVDD suffer serious disability [2,3]. The prevalence of IVDD-related lower back pain is significantly high worldwide, reducing individual living quality and increasing economic and social burdens [4,5]. Despite the considerable efforts that have been made to define potential mechanisms of disc degeneration, the etiology of its initiation and progression is still elusive, and the current treatments, conservative strategies, and surgical interventions are of limited value for IVDD [6].

Intervertebral discs function as structures bearing pressure and cushioning shock along the spine and are more severely stressed than other tissues, which makes them prone to degeneration [7]. Nucleus pulposus (NP) degeneration, annulus fibrosus (AF) rupture, and cartilage endplate calcification are reported to be involved in the occurrence and development of IVDD [8]. The initiation and progression of IVDD are associated with various factors, including RNA/DNA methylation, extracellular matrix (ECM) degradation, activated autophagy, matrix metalloproteinases (MMPs) overexpression, upregulated advanced glycation end products (AGEs), and oxidative stress induced by reactive oxygen species (ROS) [9,10].

Regulated cell death (RCD), such as apoptosis, necrosis, and pyroptosis, plays an important role in degenerative diseases and tumorigenesis [11]. Ferroptosis that is distinct from apoptosis or necrosis has been elaborated as a neoteric form of RCD and is characterized by iron-dependent lipid peroxidation of the cell membrane, labile iron deposition, excessive generation of ROS, mitochondrial shrinkage, and increased density of the mitochondrial membrane [12]. Recently, the role of ferroptosis in IVDD has attracted more and more attention from researchers [13,14,15]. Iron is an essential component in many biological processes, but excess iron can be toxic [16]. When iron-induced ROS contributes to oxidative damage, cells or tissues could suffer ferroptosis. Previous literature has demonstrated that activated ferritinophagy is also responsible for the process of ferroptosis [17].

Ferroptosis has a significant correlation with degenerative diseases, such as cartilage degeneration and neurodegenerative diseases [18]. In addition, research in recent years showed that ferroptosis and lipid peroxidation were implicated in IVDD [19]. In 2022, Ohnishi et al. conducted a systematic review and summarized the potential molecular mechanisms for IVDD, and the authors elucidated the importance of ferroptosis in the degeneration of intervertebral discs [5]. However, a generalizable framework to interpret the mechanisms of ferroptosis in intervertebral disc cells, NP cells, AF cells, and endplate chondrocytes is lacking. In this narrative review, we summarize the mechanisms of ferroptosis in IVDD to establish potential therapeutic targets, and we reveal the limitations of recently published literature. Additionally, promising directions of future research with regard to ferroptosis in IVDD are proposed.

## 2. Ferroptosis

The term “ferroptosis” was firstly coined by Dixon et al. in 2012 [12]. As early as the 1950s, researchers revealed a type of cell death by the depletion of cysteine [20]. Furthermore, in 2002, Sagara et al. proposed the term “oxytosis” as inhibiting the cystine/glutamate antiporter system Xc^−^ (system Xc^−^) in nerve cells [21]. Accumulating evidence suggests that ferroptosis is a form of RCD with a series of unique characteristics and is implicated in numerous diseases and cancers [22]. Elucidating the specific activation and inhibition mechanisms of ferroptosis can be helpful in understanding and treating diseases.

Iron-related cell death (ferroptosis) is accompanied by redox imbalance [23]. The specific mechanisms of excessive ROS production in ferroptosis have been well identified, including iron-dependent Fenton reaction [24], mitochondria [25], and enzymes of the nicotinamide adenine dinucleotide phosphate (NADPH) oxidase family [26]. The peroxidation of polyunsaturated fatty acids (PUFAs) can be induced by the strong oxidation effect of ROS, which is the main biochemical event to trigger ferroptosis [27]. On the other hand, the antioxidant systems, glutathione (GSH)–glutathione peroxidase 4 (GPX4), tetrahydrobiopterin (BH4), and coenzyme Q10 (CoQ10), play a pivotal role in maintaining redox balance to protect cells from lipid peroxidation and ferroptosis [28,29].

## 3. Ferroptosis in Degenerative Diseases

It is of great significance to explore the roles of ferroptosis-related regulatory mechanisms in degenerative diseases for both disease cognition and treatment. With the in-depth research of ferroptosis in recent years, prominent achievements have been made in degenerative diseases [30]. A variety of degenerative diseases have been revealed to be associated with ferroptosis, including neurodegenerative diseases [18], age-related macular degeneration [16], photoreceptor and retinal degeneration [31,32], osteoarthritis [33], hepatocellular degeneration [34], and kidney degeneration [35].

In addition, numerous ferroptosis-related molecular mechanisms have been revealed. Ferroptosis inhibitor, Ferrostatin-1, was verified to be effective in alleviating degenerative process [32]. Vitamin E in combination with GPX4 could protect hepatocytes from degeneration by inhibiting deleterious lipid peroxidation and ferroptosis [34]. Ferroptosis in retinal pigment epithelial cells could be aggravated by Interferon-γ, and inhibiting ferroptosis served as a promising target for preventing macular degeneration [36]. Poly rC binding protein 1 (PCBP1), a cytosolic iron chaperone, could limit the toxicity of cytosolic iron by decreasing the production of ROS and inhibiting lipid peroxidation and ferroptosis in mouse hepatocytes, which was considered an important factor for preventing liver steatosis [37].

IVDD, a representative and common degenerative disease, has also turned out to be significantly associated with ferroptosis. A number of molecular mechanisms and signaling pathways were elucidated to define potential therapeutic targets for IVDD.

## 4. Main Components of Intervertebral Disc

### 4.1. NP Cells

NP is a jelly-like avascular tissue in the center of the intervertebral disc and mainly consists of NP cells, proteoglycans, and collagen type II [38], with a function of resisting axial compression forces along the spine [39]. NP is considered as the most significant object of the degeneration of intervertebral disc. NP cell degeneration is often accompanied by a series of pathological changes, including imbalance of ECM metabolism [40], collagen type II degradation, downregulated aggrecan [41], upregulated proinflammatory cytokine [42], and oxidative stress [10]. Zhang et al. conducted a single-cell RNA-seq analysis of human NP tissues and elucidated the involvement of ferroptosis in disc degeneration pathogenesis [13].

### 4.2. AF Cells

AF is a highly fibrous and well-organized tissue surrounding the outer region of the intervertebral disc to maintain physiological intradiscal pressure [43]. Lamella, a distinct structure of AF tissues, plays a pivotal role in confining NP tissues against lateral expansion [44]. A collagen-rich matrix contains numerous AF cells, and AF cell degeneration could compromise the integrity of intervertebral discs, contributing to mechanical imbalance and intervertebral disc herniation [45]. AF cell degeneration is prone to being traced in the initiation and progression of IVDD, and autophagy induced by oxidative stress is implicated in such pathogeneses [46]. When AF cell degeneration is presented, a variety of pathological events could occur, including decreased intervertebral disc cellularity, upregulated matrix-degrading enzymes, and inflammation [47,48].

### 4.3. Endplate Chondrocytes

Cartilage endplates that consist of chondrocytes and matrix are located at the upper and lower edges of the vertebral body and resemble thin hyaline cartilage in structure [49]. Their main functions are to distribute mechanical loads along the spine and to provide nutrients to intervertebral discs [50]. Oxidative stress, RCD, and autophagy of endplate chondrocytes were reported to be responsible for the degeneration and calcification of cartilage endplate [51,52]. Accumulating evidence has demonstrated the important role of cartilage endplate in IVDD [52,53,54].

## 5. Potential Mechanisms of Ferroptosis in IVDD

### 5.1. Iron Overload

#### 5.1.1. Iron

Iron is an essential microelement to maintain the function of iron-containing proteins, which are involved in various pathophysiological processes, such as catalyzing the formation of hydroxyl radicals, inducing cell death, and regulating DNA replication [55]. Ferrous iron (Fe^2+^) and ferric iron (Fe^3+^) are the main states of iron in mammalian cells, and Fe^2+^ can be converted to Fe^3+^ through the Fenton reaction to generate excessive ROS, which triggers oxidative stress and ferroptosis [56]. Transferrin receptor 1 (TfR1), TfR2, and divalentmetal transporter 1 (DMT1) are all responsible for taking extracellular iron into the cytoplasm, and the excess cellular iron is either stored in the ferritin heavy chain (FTH) or exported by ferroportin (FPN) [57]. The mechanisms of maintaining cellular iron homeostasis are so complex that problems in any of these iron transfer pathways are likely to disturb the balance, leading to the occurrence and progression of diseases.

The role of iron overload in disc degeneration remains controversial in the literature. Wang et al. revealed that patients with severe IVDD had a higher serum ferritin level than patients with non-severe IVDD. He established an iron-overloading mouse model and demonstrated that iron overload could promote endplate chondrocyte calcification and degeneration in a ferroptosis-related manner [15]. On the other hand, Zhang et al. found that iron deficiency of human NP cells could significantly influence the synthesis and function of iron-containing proteins, such as DNA polymerase epsilon complex (PolE), eventually leading to apoptosis of NP cells [58]. Guo et al. concluded that serum iron was negatively correlated with the degree of IVDD by a clinical study [59]. Nevertheless, Aessopos et al. suggested that the serum ferritin concentration was not responsible for intervertebral disc calcification in patients with thalassemia intermedia [60]. Notably, cellular iron homeostasis is important for the normal physiological functions of the body, and both iron overload and deficiency can have disastrous consequences for intervertebral disc cells.

#### 5.1.2. Ferritinophagy

Nuclear receptor coactivator 4 (NCOA4), a selective receptor, can specifically bind FTH and disturb intracellular iron homeostasis in a ferritinophagy manner [61]. Excessive ferritinophagy could lead to labile iron overloading [62]. Yang et al. demonstrated that oxidative stress and ferroptosis induced by tert-butyl hydroperoxide (TBHP) could decrease the proliferation and vitality of rat NP cells and AF cells. Then, they elaborated the mechanism of ferroptosis-related pathways in rat NP cells and AF cells: upregulated NCOA4 combines with FTH1 and significantly enhances ferritin degradation, increasing the concentration of cellular Fe^2+^; excessive ROS generation by the Fenton reaction ultimately leads to lipid peroxidation of the cell membrane and ferroptosis [19]. Autophagy flow of the intervertebral disc cells was somehow enhanced by TBHP, but the authors did not discuss the specific regulation mechanisms for such enhancing [19]. Previous studies have shown that ROS-related oxidative stress can trigger autophagy in rat NP cells through the MAPK pathway and can upregulate matrix metalloproteinase-3 (MMP-3) in AF cells [63,64]. Autophagy is correlated with ferroptosis in a harmonious way, and such correlation warrants further investigation in intervertebral disc cells to define potential therapeutic biomarkers for IVDD.

#### 5.1.3. Iron Exportation

FPN, a multi-transmembrane protein, is the only reported transporter in mammals to export excessive intracellular ferrous iron and is mainly responsible for maintaining cellular iron homeostasis [65]. FPN deficiency can lead to labile iron accumulation in breast cancer cells, and ferrous iron-related ROS production (Fenton reaction) eventually contributes to ferroptosis [66]. Metal-regulatory transcription factor 1 (MTF1) can bind to the gene promoters to regulate the expression of target genes, which largely depends on its translocation from the cytoplasm to the nucleus [67,68]. Lu et al. elucidated that the FPN dysfunction induced by TBHP was significantly associated with intracellular iron overload and ferroptosis in human NP cells [69]. The underlying mechanism of downregulated FPN is c-jun N-terminal kinase (JNK) activation, which can attenuate the nuclear translocation of MTF1. Hinokitiol could restore the FPN function to suppress ferroptosis of NP cells in vitro and to ameliorate IVDD in vivo [69]. JNK/MTF1/FPN can act as a ferroptosis-related pathway in intervertebral disc cells and can serve as a novel therapeutic target against IVDD.

#### 5.1.4. Neovascularization

The intervertebral disc is accepted as an avascular tissue, and the cartilage endplate provides most of its nutrition [70]. As early as the 1990s, neovascularization in herniated NP tissues was observed by some researchers [71]. Arai et al. observed vascularized granulation tissues along the tears in the extruded tissues during degenerative disc herniation [72]. Oxidation of hemoglobin followed by the formation of heme moieties (ferriporphyrin) could induce cell death by iron-related oxidative damage [73]. Shan et al. clarified that heme that extravasated from neovascularization could contribute to iron accumulation and ferroptosis in human NP cells and tissues by matrix-assisted laser desorption/ionization-time-of-flight mass spectrometry (MALDI-TOF MS) [74]. In fact, previous literature demonstrated that neovascularization plays an important role in promoting IVDD [75,76]. Thus, neovascularization-induced ferroptosis could act as a promising target for IVDD treatment, and the inhibition mechanisms of neovascularization in IVDD warrant further investigation.

### 5.2. Inflammation

Inflammation has been a hallmark of various diseases, and the pathogenesis of these diseases is associated with some pro-inflammatory cytokines, such as interleukin 6 (IL-6), IL-1β, and tumor necrosis factor-α (TNF-α) [77,78]. IL-6 can mediate the expression of GPX4 through modulating the STAT3/GPX4 signaling pathway in cardiac microvascular endothelial cells [78], and GPX4 is significantly associated with ferroptosis. In addition, both clinical and basic research demonstrated that inflammation was responsible for IVDD, but research regarding the relationship between pro-inflammatory cytokines and ferroptosis in IVDD is lacking [79,80,81]. Sheng et al. demonstrated that IL-6 could induce iron overload, lipid peroxidation, and ferroptosis in cartilage cells of human intervertebral discs [7]. The level of GPX4 was significantly reduced in the degenerative cartilages compared with normal tissues, and overexpression of miR-10a-5p could alleviate IL-6 receptor-induced cartilage ferroptosis [7]. Consequently, regulating pro-inflammatory factor receptors at the post-transcriptional level to alleviate ferroptosis in intervertebral disc cells may be a promising way to treat IVDD. Not only micro RNA, but methylation or some other epigenetic modifications are also worthy of study.

### 5.3. Inhibition Mechanisms

#### 5.3.1. GPX4

Redox balance is critical to ensure normal physiological function of cells and tissues, and GPX4 is regarded as the main component of the antiperoxidative defense to maintain redox balance [82]. In combination with GSH, GPX4 alleviates the lipid peroxidation of the cell membrane by expending intracellular peroxidation substances [83]. Previous studies have proved that downregulated GPX4 would make cells more sensitive to ferroptosis, while upregulated GPX4 would decrease the sensitivity of cells to ferroptosis [84]. Homocysteine (Hcy) is the mediate product in the methionine cycle and can be converted back to methionine with the involvement of 5-methyltetrahydrofolate, and methyl transfer happens in this metabolic cycle [85]. Hcy can also be converted to cysteine through the trans-sulfuration pathway, which is one of the main sources of GSH synthesis [86]. Intracellular Hcy metabolic disorders could contribute to cell dysfunction and even cell death. Inhibition of cystathionine β-synthase (CBS), a key enzyme in the trans-sulfuration pathway, and S-adenosyl homocysteine hydrolase (SAHH), a key enzyme in the methionine cycle, could trigger ferroptosis in cancer cells [86,87].

Zhang et al. verified that higher serum Hcy concentration was associated with IVDD by analyzing clinical data [88]. He also elucidated that excessive Hcy could aggravate ferroptosis and oxidative stress in NP cells of Sprague–Dawley rats. The upregulation of DNA methyltransferase, DNMT1, and DNMT3 induced by excessive Hcy could increase the methylation of GPX4 and downregulate its protein expression [88]. Recently, some researchers have proposed that an increased level of Hcy was not the cause of diseases but the consequence, and S-adenosylhomocysteine (SAH) might be the main pathogenesis [89]. In addition, when Hcy concentration increased in the medium of vascular smooth muscle cells, the intracellular levels of SAH, an inhibitor of methylation, were significantly increased [90].

We reasoned that the methionine metabolic cycle is implicated in IVDD with complex molecular mechanisms and regulatory networks, including methylation-related and ferroptosis-related pathways, warranting further experimental verification to help clinicians with the treatment of IVDD.

#### 5.3.2. System Xc^−^

System Xc^−^, an amino acid antiporter, is composed of cystine transporter solute carrier family 7 member 11 (SLC7A11) [91]. SLC7A11 is involved in the synthesis of the major antioxidant GSH by importing cystine and upregulating cysteine [92]. Activation transcription factor 3 (ATF3) can specifically bind to the promotor of SLC7A11 to inhibit its expression, resulting in ferroptosis of cancer cells [93]. Li et al. revealed that TBHP could induce ATF3 overexpression and ferroptosis in human NP cells. ATF3 knockdown could downregulate IL-1β and upregulate SLC7A11 to suppress ferroptosis of NP cells in vitro and could alleviate the progression of IVDD in vivo [94]. miR-874-3p was elaborated to be the upstream regulator of ATF3 by binding to its target transcripts to downregulate ATF3. Notably, miRNA plays a pivotal role in mediating the expression of ferroptosis-related genes, such as System Xc^−^ and IL-6 receptor, by post-transcriptional regulation and is the key target for future research regarding ferroptosis in IVDD.

#### 5.3.3. NRF2

Nuclear factor E2-related factor 2 (NRF2) is a major regulator of antioxidant and cellular protective pathways, and both GSH-GPX4 and System Xc^−^ are its downstream targets for maintaining redox balance in cells [95]. Harada et al. showed that NRF2 could regulate intracellular iron metabolism and was associated with oxidative stress and ferroptosis in macrophages [96]. Existing evidence has demonstrated that circular RNA, a non-coding RNA, can bind to micro RNA to regulate the expression of its target genes [97,98]. Yu et al. suggested that circ_0072464 in bone marrow mesenchymal stem cell (BMSC)-derived extracellular vesicles could increase NRF2 synthesis and alleviate ferroptosis in mouse NP cells by binding to miR-431 [14]. Encouragingly, Ukeba et al. demonstrated that BMSCs co-cultured with rabbit NP cells could increase the NP cell viability and upregulate ECM [99]. Stem cells provide a potential therapeutic target against IVDD, and the specific pathways of the stem cell involved in alleviating IVDD need further verification.

## 6. Conclusions and Future Directions

Ferroptosis, an iron-dependent RCD, has been demonstrated to be implicated in the initiation and progression of various diseases since it was proposed in 2012 [12]. Recently, researchers focused on the relationship between ferroptosis and disc degeneration, and some interesting findings were revealed. BMSC-derived exosomes, micro RNAs, DNA methylation, inflammation, heme-induced iron overload, as well as methionine metabolism are involved in the ferroptosis of intervertebral disc cells and hold potential for IVDD treatment (Table 1).

Although several ferroptosis-related pathways have been identified in intervertebral disc cells, the research on ferroptosis in IVDD is currently in the preliminary stage. First, most research on ferroptosis in IVDD is conducted in NP cells (Figure 1), whereas AF cells and endplate chondrocytes, important components of intervertebral discs and promising objects for the treatment of disc degeneration [100], are not adequately studied (Figure 2). Second, as it stands, almost all the contributors to the research regarding ferroptosis in intervertebral disc cells are from China, and publication bias should not be ignored. We encourage researchers from other regions to contribute their scientific findings in this research field. Third, the participation of BH4 and CoQ10 in disc degeneration, as classic ferroptosis inhibition systems [101,102,103], also needs further verification. Fourth, the specific mechanisms of the methionine cycle in the ferroptosis of intervertebral disc cells is still unclear, and the controversy and knowledge gaps in the literature should be taken seriously [88,89,91]. Further studies are needed to confirm the role of SAH in the progression of IVDD and its specific regulatory network, such as RNA methylation. Fifth, thioredoxin reductase 1 (TXNRD1), a key regulator involved in ferroptosis induced by cysteine depletion, and PUFAs deserve further investigation to identify potential therapeutic targets for IVDD [104,105]. Sixth, as reported previously, molecular mechanisms at the post-transcriptional level are important in regulating cells ferroptosis [7,14,94]. So far, only a few miRNAs have been found to be associated with ferroptosis in intervertebral disc cells; gene and transcript methylation may be crucial factors in regulating ferroptosis and are worthy of study to identify effective inhibitors for IVDD. Consequently, ferroptosis-related signaling pathways warrant further investigation and hold potential for IVDD therapy.

## Figures and Tables

**Figure 1 cells-11-03508-f001:**
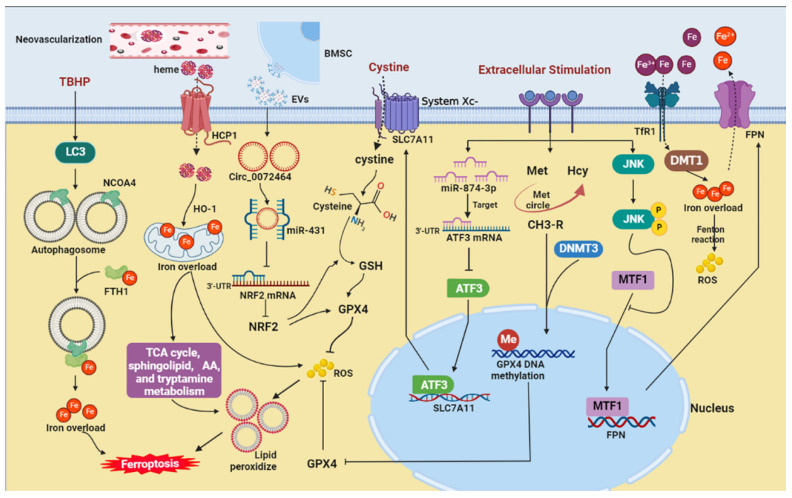
The potential ferroptosis-related pathways in nucleus pulposus cells. AA, arachidonic acid; ATF3, activation transcription factor 3; BMSC, bone marrow mesenchymal stem cell; DMT1, divalent metal transporter 1; DNMT3, DNA methyltransferase 3; EVs, extracellular vesicles; FTH1, ferritin heavy chain 1; FPN, ferroportin; GPX4, glutathione peroxidase 4; GSH, glutathione; HCP1, heme carrier protein 1; Hcy, homocysteine; HO-1, heme oxygenase-1; JNK, c-jun N-terminal kinase; LC3, microtubule-associated protein light chain 3; Met, methionine; MTF1, metal-regulatory transcription factor 1; NCOA4, nuclear receptor coactivator 4; NRF2, nuclear factor E2-related factor 2; ROS, reactive oxygen species; SLC7A11, cystine/glutamate antiporter; TBHP, tert-butyl hydroperoxide; TCA, tricarboxylic acid; TfR1, transferrin receptor 1.

**Figure 2 cells-11-03508-f002:**
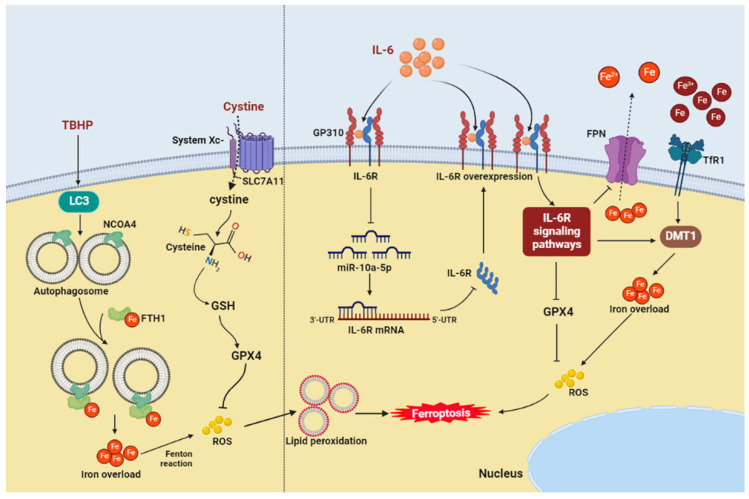
The potential ferroptosis-related pathways in annulus fibrosus cells (**left**) and endplate chondrocytes (**right**). DMT1, divalent metal transporter 1; FTH1, ferritin heavy chain 1; FPN, ferroportin; GPX4, glutathione peroxidase 4; GSH, glutathione; IL-6, interleukin-6; IL-6R, interleukin-6 receptor; LC3, microtubule-associated protein light chain 3; NCOA4, nuclear receptor coactivator 4; ROS, reactive oxygen species; SLC7A11, cystine/glutamate antiporter; TBHP, tert-butyl hydroperoxide; TfR1, transferrin receptor 1.

**Table 1 cells-11-03508-t001:** Ferroptosis in intervertebral disc cells.

Cell type	Sample	Upstream Regulator	Target	Effect	Reference
Nucleus pulposus cell	Male Sprague–Dawley rats	DNMTs	Downregulating GPX4	Inducing ferroptosis	[88]
	Sprague–Dawley rats	NCOA4	Consuming FTH1	Inducing ferroptosis	[19]
	Human	JNK/MTF1	Downregulating FPN	Inducing ferroptosis	[69]
	Human and male Sprague–Dawley rats	FTL and HO-1	Iron overload	Inducing ferroptosis	[13]
	Human	Heme catabolism	Consuming GPX4	Inducing ferroptosis	[74]
	C57BL/6 male mice	circ_0072464/miR-431	Upregulating NRF2	Inhibiting ferroptosis	[14]
	Human	miR-874-3p/ATF3	Downregulating SLC7A11	Inducing ferroptosis	[94]
Annulus fibrosus cell	Sprague–Dawley rats	NCOA4	Consuming FTH1	Inducing ferroptosis	[19]
Endplate chondrocyte	Human	miR-10a-5p	IL-6 receptor	Inhibiting ferroptosis	[7]
	C57BL/6 male mice	Ferric ammonium citrate	Accumulating ROS	Inducing ferroptosis	[15]

## Data Availability

Not applicable.

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
