# Peer review of "Targeting Ferroptosis Holds Potential for Intervertebral Disc Degeneration Therapy"

_cells, 2022, doi:10.3390/cells11213508_

Round 1

Reviewer 1 Report

This review by the Drs. Chen et al. discusses the potential of targeting the ferroptosis in the treatment of intervertebral disc degeneration (IVDD). Overall it is a well-written review on an interesting and understudied subject. I have just couple of minor comments:

1.       I think paraphrasing the title to “Targeting ferroptosis holds a great potential…”

2.       On p.2”increased ROS production” is used in the sense of “oxidative stress”. While the first is a physiological process it is in any means equal to latter (pathological process).

3.       p.2 and throughout the text “ROS accumulation” have been used –this is quite misleading as the ROS are very short lived “molecules”, main feature of which is their enormous chemical activity. Thus, while ROS cannot physically accumulate, the oxidation damage can. Please revise throughout the text.

4.       p.2 Literature search protocol –this paragraph is not necessary-please omit from the manuscript body

5.       p.2 “Ferroptosis is a kind of cell death”, better to use “a type of cell death”

6.       p.3 alleviating “degenerating “ process, please change to “degenerative”

7.       p.4 “ ROS accumulation”, see point 3

8.       p.6 “…NRF2 is a transcription factor is one of the antioxidant defence systems” please change to “ is a major regulator of antioxidant and cellular protective pathways” or similar

9.       Abstract: “Furthermore, we propose the deficiency of the published literature…” –please remove this sentence from the abstract.

10.   Table 1. At AF cells there is a random bullet point-please omit

Reviewer 2 Report

In the submitted manuscript entitled “Ferroptosis Holds a Great Potential for Intervertebral Disc Degeneration Therapy,” the authors summarized the literature regarding the relationship between intervertebral disc degeneration (IVDD) and ferroptosis. Although the authors summarised the literature, a similar topic is reviewed recently.

o   Interestingly, most studies focus on the nucleus pulposus (NP), and very few mention annulus fibrosus (AF) and cartilage endplate, which could be crucial to target IVDD.  

o   Ferroptosis is usually accompanied by metabolic imbalances of iron, lipids, and glucose and is also modulated by transcriptional and epigenetic regulation.

o   To develop effective treatment options for intervention in IVDD, the specific mechanism, molecular targets, and associated signaling pathways of ferroptosis should be established. By reading the manuscript, it seems, still, those parts are missing

o   Are any clinical drug translations for IVDD treatment

o   Authors mentioned that “BMSC-derived exosomes, micro RNAs, DNA methylation, inflammation, heme-induced iron overload, as well as methionine metabolism are involved in the ferroptosis of intervertebral disc cells and serve as potential biomarkers for IVDD treatment” What author mean by serving as potential biomarkers here?

o   The epidemiological assessment with global perspective introduction part is missing. Authors should co-relate the significance of IVDD treatment with the latest epidemiological data.

o   Authors have shown ferroptosis inhibition strategy by targeting system Xc-,Nrf2 and GPX4. However, there are other ways to inhibit ferroptosis,  like inhibition of ROS production by PUFA inhibition or inhibiting cysteine metabolism by inhibiting TXNRD1. Authors should add those approaches too.

o   Image quality is poor. The authors did not use contrasting colors between text and image, leading to difficulty reading in some parts. Please improve quality

o   Please check the manuscript for typographical, punctuation, and grammatical mistakes. There are numerous formatting issues
